psychology

preprints, credibility, trust

**Author for correspondence:**
Courtney K. Soderberg
e-mail: courtney@cos.io

# Credibility of preprints: an interdisciplinary survey of researchers

Courtney K. Soderberg[1], Timothy M. Errington[1] and Brian A. Nosek[1,2]

[1]Center for Open Science, Charlottesville VA, USA
[2]Department of Psychology, University of Virginia, Charlottesville VA, USA

(iD) CKS, 0000-0003-1227-7042; TME, 0000-0002-4959-5143; BAN, 0000-0001-6797-5476

Preprints increase accessibility and can speed scholarly communication if researchers view them as credible enough to read and use. Preprint services do not provide the heuristic cues of a journal's reputation, selection, and peer-review processes that, regardless of their flaws, are often used as a guide for deciding what to read. We conducted a survey of 3759 researchers across a wide range of disciplines to determine the importance of different cues for assessing the credibility of individual preprints and preprint services. We found that cues related to information about open science content and independent verification of author claims were rated as highly important for judging preprint credibility, and peer views and author information were rated as less important. As of early 2020, very few preprint services display any of the most important cues. By adding such cues, services may be able to help researchers better assess the credibility of preprints, enabling scholars to more confidently use preprints, thereby accelerating scientific communication and discovery.

## 1. Introduction

Scientific outputs have been growing at a rapid rate since the end of World War II; a recent estimate suggested the growth rate has been doubling approximately every 9 years [1]. Researchers are faced with far more available scholarship than they have time to read and evaluate. How do they decide what is credible, or at least credible enough to invest time to read more closely?

Historically, journal reputation and peer-review have been important signals of credibility and for deciding to read an article. Over the last 30 years, preprints (manuscripts publically shared before peer-review) have been gaining in popularity, and are promoted as a method to increase the pace of scholarly communication [2]. They are also a method for assuring open

access to scholarly content as virtually all preprint services offer 'green open access'—openly accessible manuscript versions of papers whether or not they are published. However, preprints lack the journal-centric cues for signalling credibility. So how do researchers decide whether preprints are credible and worth reading? We conducted a survey to gather information about cues that could be displayed on preprints to help researchers assess their credibility. Understanding whether and how heuristic cues can be used for initial assessments of credibility is important for meeting the promise that preprints hold for opening and accelerating scholarly communication.

## 1.1. Credibility and preprints

The informal sharing of pre-peer-review or pre-publication drafts has long been a part of scientific discourse (called 'preprints', 'working papers', or 'manuscript drafts' depending on the discipline—here we refer to these all as 'preprints', using the emerging standard term). Mechanisms for more formal dissemination emerged in the early 1990s with arXiv, a repository that now hosts more than 1.3 million preprints in physics, mathematics, and allied fields. SSRN, a preprint service originally for social science research, started in 1994. And, since 2013, more than two dozen preprint services have launched representing a wide variety of topics, indicating growing recognition of this mechanism of communication across all areas of scholarship.

Preprints can speed the dissemination of research [2–4]. Because preprints are not part of the often-long peer-review journal process, researchers can make their findings available sooner, potentially spurring new work and discoveries by others who read the preprint. However, preprints can only accelerate scientific discovery if others are willing to judge them as credible, i.e. legitimate research outputs that they can read, cite, and build upon. There exists some skepticism about the credibility of preprints, particularly in those fields for which the concept is new. A large international survey conducted in the early 2010s found that researchers from a broad variety of disciplines reported that citing non peer-reviewed sources or preprints was not very characteristic of citation practices in their disciplines [5]; NIH only changed their policy to allow preprints to be cited in grant applications in March of 2017; and some journals only very recently allowed preprints to be cited in articles [6]. Articles initially posted as preprints on bioRxiv have a citation advantage, but this may come from posting the research on a freely available platform, thus expanding access to those unable to read behind a paywall, and not from the citation of the preprint during its pre-publication period [7].

Why do researchers rarely cite preprints? Focus groups [8] and large multinational surveys [5] found that reputation, of both journals and articles, was important for researchers when determining what to read and use. Reputation was often defined by more traditional citation-based metrics, including journal impact factor and author $h$-index [8]. Traditional peer-review was also singled out as particularly important. In the previous survey [5], scholarly peer-review was rated as the most important factor for determining the quality and trustworthiness of research and was important for determining what to read and use. Peer-review and journal reputational cues are lacking from preprints. If these are important pre-conditions for researchers to engage with scholarly work, preprints will be disadvantaged when compared with published work.

## 1.2. Potential cues for initial credibility assessments of preprints

Reading scholarly works is an important part of how researchers keep up with the emerging evidence in their field, and explore new ideas that might inform their research. In an information-rich environment, researchers have to make decisions about how to invest their time. Effective filters help researchers make decisions about continuing with deeper review or stopping and moving on. For example, on substance, an effective title will provide the reader with cues about whether the topic of the paper is relevant to them. If it is, an effective abstract will help the reader determine whether it is not relevant after all, that the paper is worth reading in full, or that the gist from the abstract is enough.

Assessing substantive relevance is easier than assessing whether the paper meets quality standards that make it worth reading and using. Journal reputation and peer-review are methods of signalling that others independently assessed the work and deemed it worthy enough to publish. Even though previous research has shown that the peer-review process is unreliable (e.g. low test–retest reliability of article acceptance [9], reviewers catch a small proportion of major mistakes in submissions [10], strong confirmation biases of reviewers [11], and poor inter–rater reliability between reviewers and between editors and reviewers [12]), the signalling function is highly attractive to researchers that need some cues to help filter an overwhelming literature with their limited time.

Without journal reputation and peer-review, are there any useful cues that can be provided to help researchers filter preprints based on quality or credibility? To date, no work that we know of has investigated credibility cues on preprints specifically. There are, however, models that propose heuristics for how people judge the credibility of online information broadly [13–16]. Previous work on credibility judgements of scholarly work mostly assessed cues that were already present, rather than investigating potential new cues. For the present work, we sampled cues that could be helpful for assessing credibility, whether they currently existed or not.

### 1.2.1. Cues for openness and best practices

Transparency, openness, and reproducibility are seen as important features of high-quality research [17,18] and many scientists view these as disciplinary norms and values [19]. However, information about the openness of research data, materials, code or pre-registrations, either in preprints or published articles, is often not made clear. Previous work by the Center for Open Science (COS) has led to over 60 journals adopting badges on published articles to indicate open data, materials, and/or pre-registrations, and such signals could also be adopted by preprint services [20,21].

Such information could match up well with the more systematic, thorough processes researchers go through when assessing the credibility of articles. Previous work [5] found that many researchers rate determining whether the 'data presented in the paper are credible' as very important for use/reading decisions. A recent Pew Research survey [22] found that a majority of U.S. adults trust scientific findings more if the data is made publically available. Additionally, work by Piwowar & Vision [23] suggested that an increase in citations for articles which shared data could be due at least in part to data sharing signalling credibility of the work. To the extent that communities value certain research behaviours (e.g. data sharing, code sharing, pre-registration), providing cues that indicate a preprint engages in these behaviours could signal the credibility of the preprint.

### 1.2.2. Cues of others' evaluation and interest

Work on credibility judgements of internet content identified a number of heuristics that rely on cues about the opinions of others: bandwagon/consensus heuristic (if many others [known or anonymous] think it is good, so should I; [14,24]), endorsement (trust sites and sources that are recommended by others [16]), and liking/agreement heuristic (tending to agree with the opinions of others I like [24]). Peer-review may function as this type of heuristic cue, and so displaying information such as download or view counts, endorsements, or community comments on preprints could serve a similar purpose. Previous work found that displaying download count information about papers alongside abstracts can alter download behaviours [25]. Though downloads are not a direct measure of credibility, they are moderately correlated with citations [26,27] and past research has used them as an indication of user trust and satisfaction [28,29].

### 1.2.3. Cues of the reputation of authors or research origins

Authority [14] and source reputational heuristics [16] suggest a number of cues that might be added to preprints. Preprints do not have journal reputation to rely on for credibility cues, but adding information about authors (e.g. author institutions, verified identities markers like ORCID digital identifiers, links to Google Scholar pages) could increase the extent to which author reputational cues are used [30,31]. Cues about conflicts of interest of authors or funders, which may carry their own reputational clout, could also tap into these heuristics.

### 1.2.4. Cues of independent verification

Consistency heuristics (a piece of information should be judged as credible when it is found to agree with information from other independent sources [16]) suggest other types of useful cues. If preprint services could signal the extent to which others could verify author claims (e.g. that data is actually available, that results are robust to other analysis choices), these cues could affect the credibility judgements of individual preprints.

### 1.2.5. Variability in cue use

Previous research indicates that the cues mentioned above are used generally for making credibility judgements, but there is variability in the relative importance placed on different types of cues. For example, when judging the credibility of scholarly work, all research disciplines strongly endorsed the trustworthiness of the peer-review process. But, life-scientists endorsed peer-review as an indicator of credibility more strongly than respondents from other disciplines, and social scientists more strongly endorsed recommendations by colleagues [5]. There were also differences by the age [32] and Human Development Index (HDI) of the countries researchers were in [33]. This suggests that different types of researchers might find different heuristic cues more relevant or salient than others when assessing credibility.

## 1.3. The present research

In the spring of 2019, we conducted a survey of active researchers. We assessed the extent to which cues are considered important for credibility judgements about preprints, and how this varied across disciplines and career stages.

# 2. Results

In total, 4325 researchers consented to take the survey. Of those, 13.09% answered no questions after consent leaving us with a sample of 3759 respondents. Of those, 13.25% dropped out part-way through the survey. All questions in the survey were optional, so even those who completed the survey did not necessarily have complete data. We retained respondents who only partially completed the survey. We cannot calculate an overall response rate to the survey because we cannot determine the total number of people contacted with the diverse outreach methods. All analyses reported below are exploratory, as specified in our pre-registration (https://osf.io/wbpxy) [34], and the data underlying these analyses can be found at https://osf.io/j7u6z/ [35].

## 2.1. Participant characteristics

Overall the sample was quite familiar with preprints, with 52.83% of the sample saying they were very or extremely familiar with preprints, and only 5.43% reporting that they were not at all familiar with preprints. The sample was largely balanced in terms of academic career stage, with 33.97% of the sample being graduate students or post docs, 33.28% being an assistant, associate, or full professor and 32.75% either not answering the question or not falling into one of the previously listed career stages. In terms of discipline, the four largest categories, based on the bepress taxonomy, were the social sciences (35.20%), life sciences (24.10%), physical science and mathematics (10.90%), and medical and health sciences (7.95%). Within social scientists, 67.07% self-identified as psychologists. Of respondents, 15.30% either did not report this discipline or did not report something that could be clearly categorized. Finally, the majority of the sample was from North America (31.7%) or Europe (35.3%), with US researchers specifically making up 26.96% of the total sample. Of respondents, 15.8% did not list their country. The vast majority of respondents, 71.70%, also came from countries with very high HDI scores. We had hoped to analyse geographical differences, but unfortunately, we felt that our sample was far too skewed towards very high/high HDI and United States and Western European samples for us to draw any meaningful conclusions based on differences we might see in our data.

## 2.2. Engagement with preprints

### 2.2.1. Favourability towards preprints

The sample was mostly favourable towards preprints; 69.73% of the sample felt slightly to strongly favourable towards preprints, while only 15.16% felt opposed to preprints and 14.95% felt neutral. On average, all disciplines and career stages felt favourable towards preprints, though there were slight differences in the extent of this favourability. Psychology and other social science disciplines showed the highest level of favourability, with engineering, biology, and physical and mathematical sciences showing only slightly lower favourability (figure 1). Only one discipline, medicine, had fewer than 60% of respondents favour the use of preprints, and even there, the majority of respondents (51%)

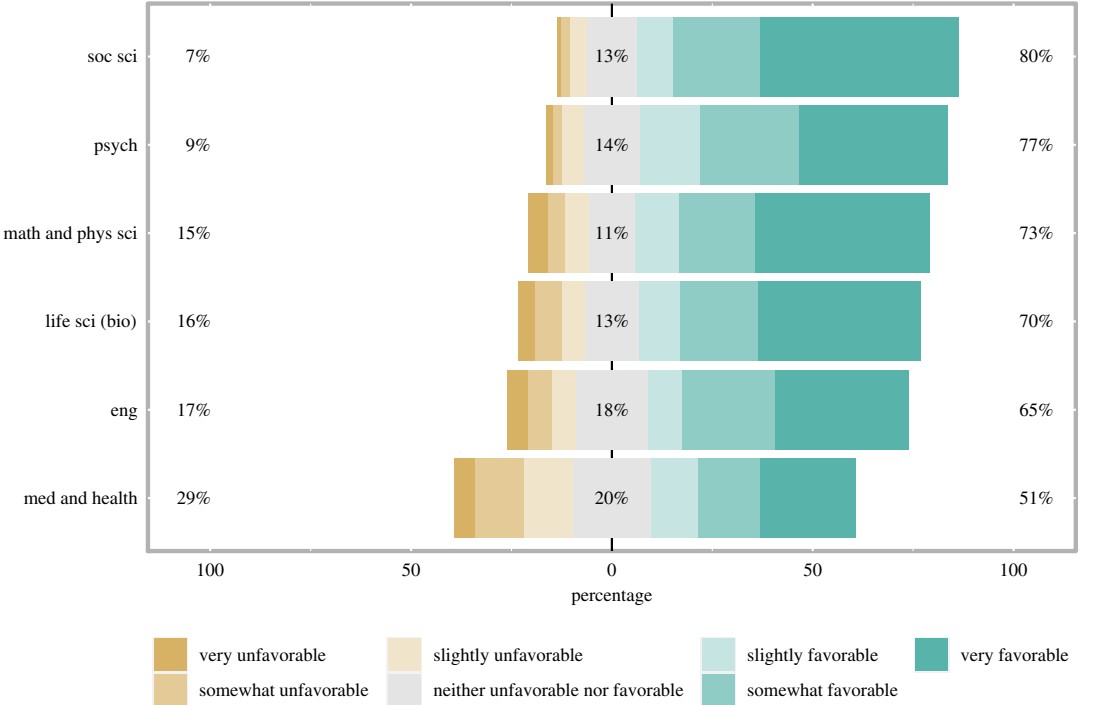

**Figure 1.** Favourability towards preprints by discipline. Respondents favourability towards the use of preprints in their discipline, broken out by the six most common disciplines in our sample. Numbers to the left of the bars indicate the percentage of respondents who responded with 'very unfavorable', 'somewhat unfavorable' or 'slightly unfavorable', the numbers in the centre of the bars indicate the percentage who responded, 'neither unfavorable nor favorable', and the numbers to the right of the bars indicate the percentage who responded, 'very favorable', 'somewhat favorable' or 'slightly favorable'.

reported feeling favourable. Among careers stages, graduate students (80%) and postdocs (78%) showed the highest level of favourability while full professors (61%) showed the lowest levels (figure 2).

### 2.2.2. Use of preprints

Of respondents, 72.92% had either viewed/downloaded or submitted preprints at least a few times. However, repeated usage was associated with preprint viewers/downloaders more than by preprint submissions. Of respondents, 70.63% had viewed/downloaded preprints either a few or many times, while only 29.85% had submitted a preprint a few or many times.

Among disciplines, medicine had by far the lowest levels of viewing/downloading of any discipline (figure 3*a*). The life sciences and psychology showed slightly higher percentages of respondents who had either never viewed/downloaded a preprint or only done it a few times as opposed to many times. This may reflect the fact that dedicated preprint services in these disciplines are newer than in other disciplines, and so they have had less time to develop norms around such behaviour. Preprint submission was much lower overall than rates of reading/citing; across all disciplines, the most common response category was that respondents had never submitted a preprint (figure 3*b*). Even in the physical sciences and mathematics, which have the longest history with preprint services, rates of submission were low.

The use of preprints varied by academic career stage, though not in a clear linear pattern. Though postdocs showed the highest levels of viewing/downloading, and full professors the lowest, graduate students and associated professors were not markedly different (figure 4*a*). As with discipline, the rate of preprint submissions was quite low (figure 4*b*). Graduate students and postdocs showed the lowest levels of submission, perhaps because they have had fewer opportunities to post/have their work posted as preprints. Professors of all levels were more likely to have submitted a preprint.

## 2.3. Credibility of preprints

Of the 19 questions on what information would be important when judging the credibility of a preprint, eight items were rated as either very or extremely important by a majority of respondents (figure 5). With

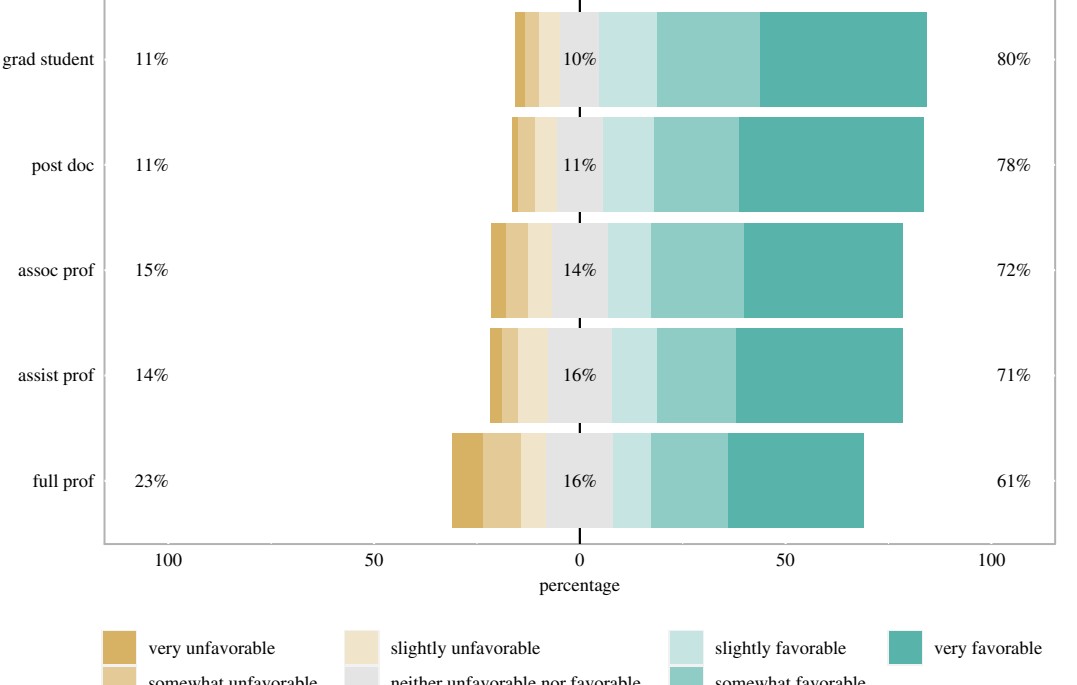

**Figure 2.** Favourability towards preprints by academic career stage. Respondents favourability towards the use of preprints in their discipline, broken out by academic career stage. Numbers to the left of the bars indicates the percentage of respondents who responded with 'very unfavourable', 'somewhat unfavourable', or 'slightly unfavourable', the numbers in the centre of the bars indicates the percentage who responded, 'neither unfavourable nor favourable', and the numbers to the right of the bars indicates the percentage who responded, 'very favorable', 'somewhat favorable', or 'slightly favorable'.

the exception of COIs, these eight cues are related to indicators of transparency/openness of research content and process (e.g. links to data, links to pre-analysis plans) or to verification of author claims by independent groups (e.g. computational reproducibility, accessing linked information). Only two cues were rated as not at all or slightly important by a majority of respondents (i.e. simplified endorsements and anonymous comments). In general, information related to field perceptions/usage of preprints was rated as less important by respondents, and information related to preprint authors (e.g. institutional information, previous work) received more muted support.

### 2.3.1. Relationship with use and favourability towards preprints

To increase the credibility of preprints, it may be particularly useful to know what items were rated as important for judging credibility by respondents who use preprints less or view them less favourably. To investigate this, we ran correlations between participants' ratings of the 19 potential cues and the extent to which respondents, or their co-authors, had submitted preprints; the extent to which they view/download preprints (both converted into ordinal variables, with 1 representing 'never', and 4 coded as 'yes, many times'); and the extent to which they favoured the use of preprints. The results can be seen in figure 6. We used Spearman correlations for the view/download and submit correlations due to the ordinal nature of the data and a Pearson correlation for favourability.

Figure 6 shows that most of the correlations are small (median absolute value $r = 0.064$). The largest correlations involved information about a preprint having been submitted to a journal, with those that feel less favourability towards preprints ($r = -0.26$) and use them less ($r = -0.20$) rating this as more important for making credibility judgements. In general, it appears that more 'traditional' metrics (e.g. author information, peer-review information) are negatively correlated with preprint views/ downloads and favourability, indicating that these items were seen as more important by those lower on these scales, while the open science and independent verification indicators tended to be rated as more important by those who tended to favour and use preprints. In general, the preprint submission variable did not correlate as strongly with the open science or independent verification indicators as either preprint viewing/downloading behaviour or favourability towards preprints.

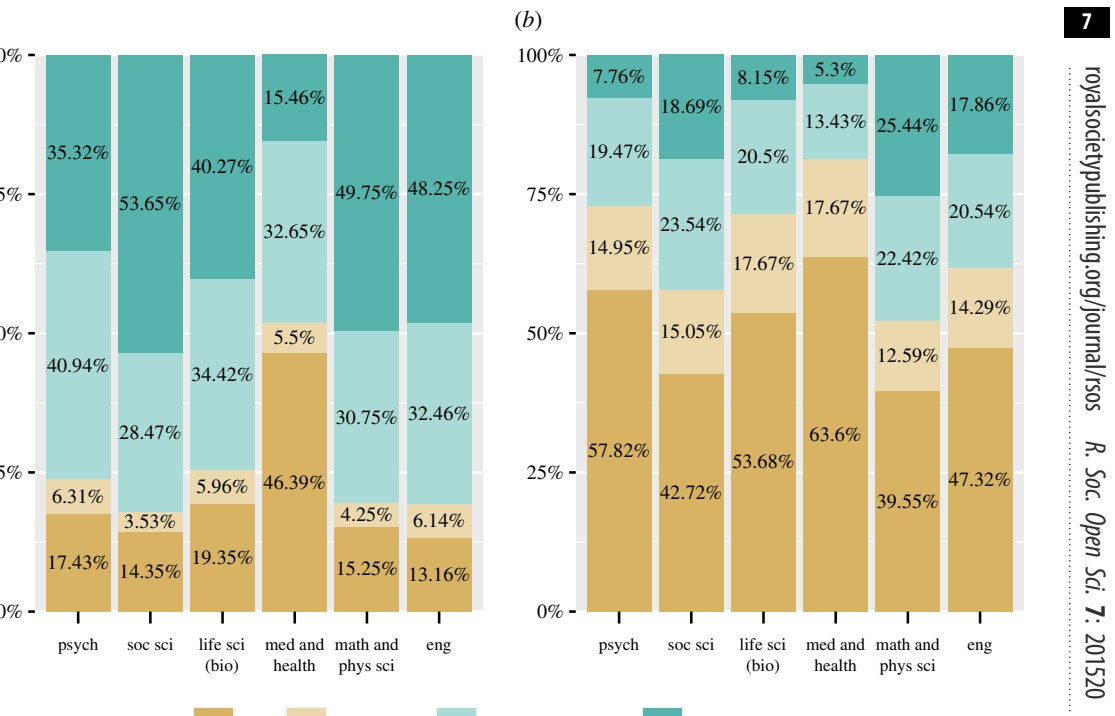

**Figure 3.** Preprint use by discipline. Whether respondents had ever 'viewed/downloaded a preprint' (*a*) and whether they or a co-authors had 'submitted a preprint' (*b*), broken up by discipline. Respondents who did not answer the question or who answered 'not sure', are not included in the graphs.

### 2.3.2. Relationship between discipline and career stage

We investigated the extent to which the importance of cues varied across discipline and academic career stage [5,32]. Figure 7 shows the breakdown of the importance of cues by discipline[1]. Though there are small differences between disciplines (e.g. psychology rates author's previous work and institutions lower than other disciplines; biology rates comments higher than other disciplines), there is consistency across disciplines on the rank ordering of average importance ratings. Differences between cues are bigger than differences between disciplines. To quantify the difference in variance in importance ratings explained by discipline and cue question, we ran a mixed-model, with discipline, cue, and their interactions as fixed effects and a random effect for participants and calculated the $R^2$ for each fixed effect [36]. In the model, cues explained 10.13% of the overall variation in importance ratings while discipline explained only 1.29% of the variation. In terms of the magnitude of the differences among cues and disciplines, the largest mean differences (on a 1-to-5 point Likert scale) between disciplines on any given question was 0.67, while the largest difference between questions for any given discipline was 1.82.

Similarly, the mean differences in item importance between career stages (e.g. graduate students and postdocs rating author information more negatively than full professors) are quite small (figure 8). We again ran a mixed-model with career stage, cue, and their interaction as fixed effects and participant as a random effect. We found that cues explain 18.90% of the variation in important ratings, and career stage explained only 1.36% of the variation. The maximum difference between the mean for any given question across career stages was 0.51, while the maximum mean difference (on a 1-to-5 Likert scale) between questions in any given career stage was 1.71. Thus, across both discipline and career stage, cues appear to be more important for driving importance ratings than participants' discipline or career stage.

### 2.3.3. Factor structure of items

If certain cues are judged similarly by respondents, this could reflect classes of indicators, with specific indicators within each class perhaps being psychologically interchangeable. Space on preprint landing

---

[1]For simplicity, we excluded some small disciplines from the analysis. These bepress tier 1 disciplines were business, education, engineering, and arts and humanities.

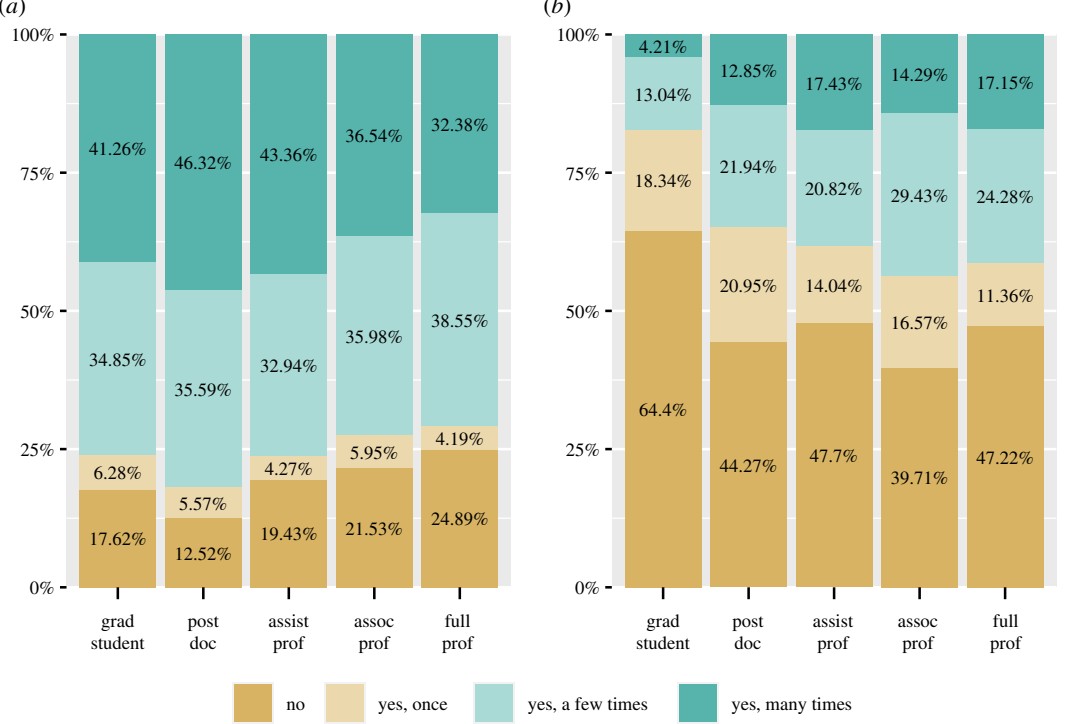

**Figure 4.** Preprint use by academic career stage. Whether respondents had ever 'viewed/downloaded a preprint' (*a*) and whether they or a co-author had 'submitted a preprint' (*b*), broken up by career stage. Respondents who did not answer the question or who answered 'not sure' are not included in the graphs.

pages is not infinite, and so understanding how researchers cluster various indicators could help services prioritize which information to show to most efficiently use presentation space. Additionally, for services that can display more indicators, understanding which cues provide similar types of information would allow services to group psychologically related cues. Grouping related cues can increase the salience of that class of information thus making it more likely to be used in making judgements [37].

We used exploratory factor analysis (EFA) to investigate potential factor structures of the 19 items. A parallel analysis suggested that the data contained six factors. We extracted a six-factor solution using maximum likelihood (ML) and an oblimin rotation. Overall, the fit of the model was adequate, with a TLI = 0.965 and an RMSEA of 0.041, 90% CI [0.038, 0.044]. Though four of the factors are small, with eigenvalues less than 1 and explaining less than 10% of the variance, the fit of the model rapidly grew worse with simpler factor structures. For example, a four-factor solution resulted in a TLI = 0.779 and an RMSEA = 0.103, 90% CI [0.100, 0.105]. Because of fit indices, the parallel analysis, and interpretability, we retained the six-factor solution.

A diagram of the structure can be seen in figure 9. The first two factors conceptually appear to map well onto an 'openness/transparency' and an 'independent verification' concept. The third and fourth factors contain questions related to 'peers views' and 'external support', respectively. Finally, the fifth and sixth factor contains most of the questions related to 'usage metrics' and 'author information'.[2] Based on the correlation structure of the factors, the 'openness/transparency' and 'independent verification' factors are highly related as are the 'peers views', 'usage metrics', and 'author information' factors. This may indicate that more traditional metrics are more closely associated than newer metrics.

### 2.3.4. Current use of cues on preprint services

We explored the extent to which the various cues we investigated are implemented on existing preprint services. In September 2019, we coded eight services, representing a number of disciplines and companies

---

[2]The factor structure appears to be broadly used by our respondents, as we found strong measurement invariance of the structure across discipline and career stage using the [38] criterions based on a combination of CFI and RMSEA changes. This indicates that different disciplines and career stages interpret the factors in conceptually similar ways, and the same items are grouped together across disciplines and career stages.

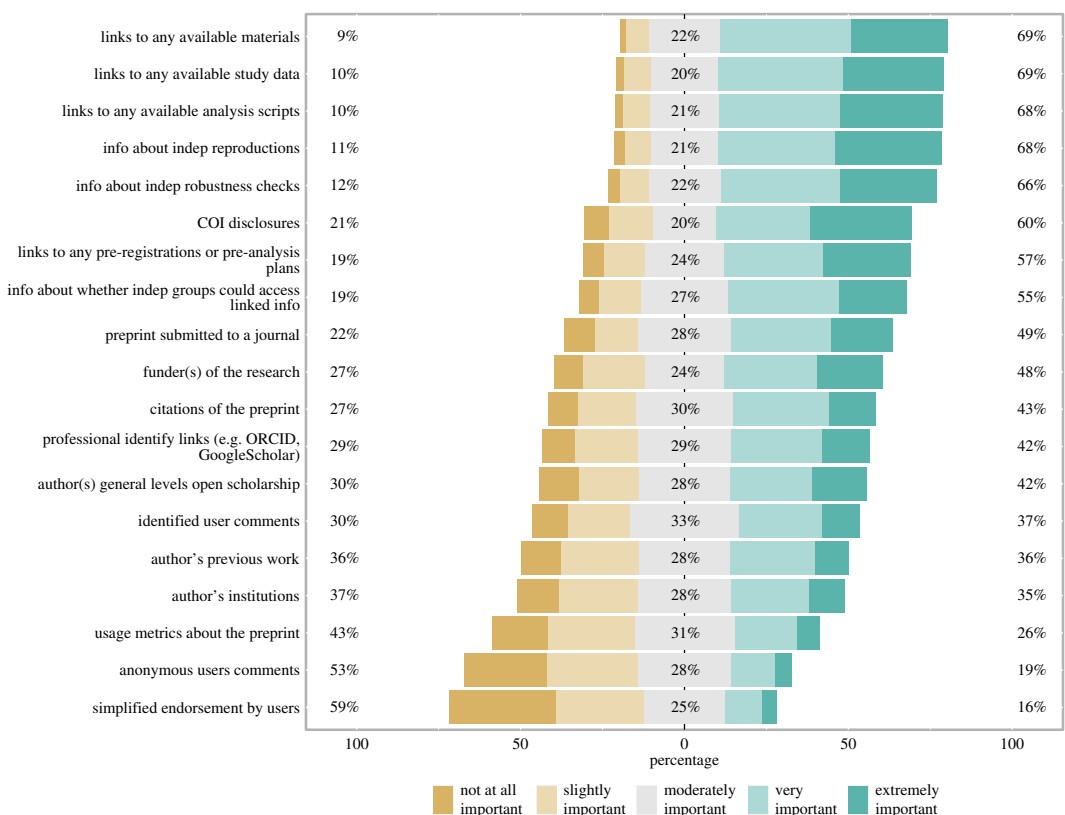

**Figure 5.** Importance of information for credibility judgements. Response to each of the 19 survey questions asking about the importance of each type of information listed for making credibility judgements about a preprint. The number to the left of the bars indicates the percentage of respondents who responded with 'not at all important' or 'slightly important', the number in the centre of the bar indicates the percentage who responded, 'moderately important', and the percentage to the right of the bar indicates the percentage who responded, 'very important' or 'extremely important''.

in the preprints space. For companies that host multiple different preprint services (e.g. Figshare and COS), we chose the first preprint service offered by each company. We coded whether each service displayed each cue on their preprint pages. To be marked 'yes', the preprint service had to meet two criteria. First, the cue had to be displayed on the page that the service controls, rather than in the uploaded preprint document itself. This avoided scoring services based on idiosyncratic features of the uploaded preprints that happened to be examined during the coding process. Second, the service needed to specifically identify the cue content. For example, as of September 2019, OSF Preprints enabled uploading of supplementary material including data and code, but the preprint pages only indicated a link to 'supplementary material' rather than specifically identifying the cue content (e.g. open data). So, OSF Preprints were coded as 'no' for data sharing cues. The results are shown in figure 10.

None of the 19 items were cued on all preprint services, and only the 'usage metrics' cue was displayed by a majority of the services. The preprint service that displayed the most cues was PeerJ (7 of the 19 cues, 36.84%). At the time of coding, this service was still active, but it stopped accepting new preprints in December 2019. Across services, many of the few cues that were present were not prominently placed, often appearing far down the page or on a separate tab. This means that even the few cues that are being displayed may not have their intended impact because users are never exposed to them or they are not very salient for users.

We observed especially poor coverage of the eight cues rated as the most important by researchers. Though many preprint services have some way of uploading 'supplementary material,' most do not differentiate between different types of information or cue particular types of content and few services enable preprint authors to provide links if relevant material (e.g. open data, pre-registrations) are stored on another service. As such, while it is possible for preprint authors to engage in the open practices that readers report as credibility enhancing, the preprint services are not yet effective at providing visual cues that those behaviours are occurring.

| | view/download preprints | submit preprints | favor use |
|---|---|---|---|
| author's previous work | −0.10 | −0.06 | −0.11 |
| author's institution | −0.10 | −0.08 | −0.10 |
| professional identity links | −0.07 | −0.05 | −0.02 |
| COI disclosures | −0.06 | −0.11 | 0.01 |
| author's level of open scholarship | −0.06 | −0.07 | 0.04 |
| funders of research | −0.10 | −0.10 | −0.00 |
| preprint submitted to a journal | −0.20 | −0.22 | −0.26 |
| usage metrics | −0.02 | 0.02 | 0.07 |
| citations of preprints | 0.01 | 0.01 | 0.10 |
| anonymous comments | −0.03 | −0.03 | 0.06 |
| identified comments | 0.03 | −0.02 | 0.12 |
| simplified endorsements | −0.05 | −0.02 | 0.04 |
| link to study data | 0.13 | 0.03 | 0.15 |
| link to study analysis scripts | 0.17 | 0.05 | 0.17 |
| link to materials | 0.11 | 0.01 | 0.13 |
| link to pre-reg | 0.06 | −0.03 | 0.11 |
| info about indep groups accessing linked info | 0.11 | 0.04 | 0.18 |
| info about indep group reproductions | 0.08 | −0.02 | 0.10 |
| info about indep robustness checks | 0.04 | −0.02 | 0.08 |

**Figure 6.** Correlations between preprint credibility questions and the extent to which participants favour the use of preprints, submit preprints and view/download them.

## 3. Discussion

We surveyed 3759 researchers about their perceptions of the importance of different cues for assessing the credibility of preprints. We observed that cues related to openness and independent verification of author assertions were rated more highly than cues related to author identities and peer-review and usage indicators. However, researchers more skeptical of preprints tended to rate author- and peer-review-related cues somewhat more highly than researchers who supported preprints. Nevertheless, there was broad agreement that transparency of the underlying research content (i.e. data, materials, code, pre-registration) and evidence of independent verification of content and research claims were the most important factors for assessing the credibility of preprints.

We observed small differences in cue ratings for both academic career stage and discipline, but the differences between career stages and disciplines were much smaller than the differences between cues. The factor structure of items was consistent across the researcher type, indicating that the cues tend to be treated similarly across discipline and career stage. In all subgroups, openness and independent verification cues had higher importance ratings than author identity and peer-review and usage cues. There may be cues that could show large disciplinary or career-stage differences—we only asked participants about a subset of all possible cues—but we did not identify them in this study. Instead, the present evidence suggests that there are a shared set of cues that can be applied across scholarly preprint communities to improve assessment of research credibility.

We also observed that existing preprint services display few of the cues that we investigated, particularly those rated as most important: openness and independent verification. This suggests that preprint services could improve support of preprint readers' assessment of research credibility by implementing some of these cues prominently with each preprint. Openness cues should be relatively

| potential icon | credibility of preprints by discipline | | | | |
|---|---|---|---|---|---|
| | psychology | soc sci | life sci (bio) | med & health sci | math & phys sci |
| | *n* = 885–888 | *n* = 431–436 | *n* = 902–905 | *n* = 296–298 | *n* = 409–411 |
| author's previous work | 2.71 (1.13) | 3.11 (1.21) | 3.03 (1.18) | 3.06 (1.19) | 3.18 (1.16) |
| author's institution | 2.67 (1.16) | 3.27 (1.13) | 2.91 (1.19) | 3.23 (1.17) | 3.01 (1.18) |
| professional identity links | 3.05 (1.13) | 3.36 (1.22) | 3.17 (1.22) | 3.34 (1.17) | 3.14 (1.20) |
| COI disclosures | 3.67 (1.23) | 3.55 (1.26) | 3.71 (1.21) | 3.99 (1.08) | 3.32 (1.38) |
| author's level of open scholarship | 3.12 (1.28) | 3.09 (1.24) | 3.20 (1.20) | 3.42 (1.18) | 2.97 (1.30) |
| funders of research | 3.35 (1.18) | 3.50 (1.21) | 3.20 (1.23) | 3.71 (1.15) | 3.07 (1.32) |
| preprint submitted to a journal | 3.37 (1.17) | 3.20 (1.24) | 3.37 (1.21) | 3.56 (1.17) | 3.40 (1.19) |
| usage metrics | 2.50 (1.10) | 2.82 (1.18) | 2.76 (1.16) | 2.98 (1.16) | 2.70 (1.17) |
| citations of preprints | 3.05 (1.12) | 3.37 (1.16) | 3.20 (1.18) | 3.37 (1.13) | 3.29 (1.16) |
| anonymous comments | 2.36 (1.12) | 2.21 (1.08) | 2.62 (1.17) | 2.56 (1.14) | 2.39 (1.18) |
| identified comments | 3.00 (1.14) | 2.88 (1.13) | 3.26 (1.15) | 3.08 (1.13) | 3.08 (1.17) |
| simplified endorsements | 2.09 (1.06) | 2.14 (1.08) | 2.40 (1.19) | 2.53 (1.21) | 2.29 (1.19) |
| link to study data | 3.81 (1.05) | 3.86 (1.02) | 3.97 (0.96) | 3.75 (1.02) | 3.91 (0.99) |
| link to study analysis scripts | 3.82 (1.06) | 3.83 (1.03) | 3.99 (0.98) | 3.74 (0.98) | 3.89 (1.00) |
| link to materials | 3.89 (0.96) | 3.77 (1.03) | 3.99 (0.94) | 3.84 (0.93) | 3.85 (0.97) |
| link to pre-reg | 3.91 (1.09) | 3.49 (1.22) | 3.44 (1.20) | 3.77 (1.03) | 3.31 (1.26) |
| info about indep groups accessing linked info | 3.44 (1.14) | 3.53 (1.12) | 3.59 (1.10) | 3.47 (1.07) | 3.49 (1.16) |
| info about indep group reproductions | 3.86 (1.04) | 3.78 (1.10) | 3.93 (1.07) | 3.82 (1.04) | 3.86 (1.03) |
| info about indep robustness checks | 3.77 (1.06) | 3.77 (1.05) | 3.90 (1.02) | 3.85 (1.05) | 3.72 (1.13) |

**Figure 7.** Responses to preprint credibility questions by disciplines. Mean and standard deviation of response to preprint credibility questions by disciplines. Respondents who either skipped the question or could not be categorized into a bepress tier 1 taxonomy are not included in the table. Additionally, participants who listed their discipline as Business, Law, Education, Engineering or Arts and Humanities were also excluded because there were too few respondents in these categories. The response scale is 1—not at all important, 2—slightly important, 3—moderately important, 4—very important, 5—extremely important.

easy to implement, but cues for independent verification are more difficult to implement for wide application because of conceptual issues in understanding verification practices, implementation challenges in creating widely applicable verification workflows, and resource challenges in conducting verification processes. Given the high value placed on these cues by the community, services and funders should work together to build capabilities to gather and display evidence of verification. This could include creating tools for the community to provide information about independent verification attempts or tools for computationally reproducible manuscripts to be uploaded to services.

Though we did not directly ask respondents about how they would rate the importance for these cues for journal articles, it is reasonable to infer journals could also benefit readers by providing cues related to openness and independent verification. Evidence from a recent Pew Research Center survey [22] found that openly available data increased trust in scientific research findings, indicating that there may be broad support for certain types of cues on papers of all types, preprints or peer-reviewed publications.

## 3.1. Limitations

Our sample is large, but it is a convenience sample recruited by marketing to a variety of research communities. As a consequence, our survey does not provide population estimates of researcher opinions, and the estimates could be biased by factors influencing self-selection to participate. Notably, our sample had decent coverage of career stage and research discipline, however, the vast majority of our sample was from the U.S. and/or Western Europe and was quite favourable towards preprints. We observed that favourability towards preprints was correlated with the ratings of certain cue items. So, to the extent that the population of researchers has lower favourability towards preprints than our sample, the overall importance placed on different cues may look different in the

| potential icon | career stage | | | | |
|---|---|---|---|---|---|
| | grad student | post doc | assist prof | assoc prof | full prof |
| | $n = 759–763$ | $n = 512–513$ | $n = 430–431$ | $n = 357–359$ | $n = 459–461$ |
| author's previous work | 2.90 (1.17) | 2.81 (1.16) | 2.92 (1.21) | 2.98 (1.15) | 3.20 (1.22) |
| author's institution | 2.84 (1.22) | 2.82 (1.14) | 2.90 (1.15) | 2.93 (1.17) | 3.01 (1.23) |
| professional identity links | 3.17 (1.16) | 3.06 (1.17) | 3.19 (1.25) | 3.19 (1.14) | 3.18 (1.24) |
| COI disclosures | 3.78 (1.22) | 3.61 (1.23) | 3.62 (1.21) | 3.50 (1.26) | 3.59 (1.33) |
| author's level of open scholarship | 3.30 (1.22) | 3.01 (1.22) | 3.09 (1.24) | 3.01 (1.25) | 3.03 (1.31) |
| funders of research | 3.36 (1.21) | 3.27 (1.21) | 3.33 (1.23) | 3.23 (1.22) | 3.23 (1.29) |
| preprint submitted to a journal | 3.39 (1.15) | 3.27 (1.19) | 3.33 (1.25) | 3.35 (1.22) | 3.41 (1.22) |
| usage metrics | 2.77 (1.13) | 2.67 (1.10) | 2.64 (1.20) | 2.66 (1.17) | 2.55 (1.15) |
| citations of preprints | 3.34 (1.13) | 3.23 (1.11) | 3.10 (1.20) | 3.09 (1.15) | 2.93 (1.24) |
| anonymous comments | 2.50 (1.19) | 2.54 (1.17) | 2.42 (1.18) | 2.41 (1.07) | 2.22 (1.12) |
| identified comments | 3.10 (1.15) | 3.19 (1.15) | 3.03 (1.18) | 3.02 (1.13) | 2.89 (1.17) |
| simplified endorsements | 2.31 (1.19) | 2.31 (1.13) | 2.23 (1.16) | 2.21 (1.09) | 2.09 (1.12) |
| link to study data | 3.99 (0.94) | 3.88 (1.02) | 3.81 (1.04) | 3.83 (0.99) | 3.70 (1.08) |
| link to study analysis scripts | 4.02 (0.97) | 3.95 (0.99) | 3.84 (1.02) | 3.81 (1.02) | 3.65 (1.09) |
| link to materials | 4.02 (0.90) | 3.92 (0.97) | 3.85 (0.98) | 3.84 (0.96) | 3.74 (1.05) |
| link to pre-reg | 3.82 (1.11) | 3.65 (1.15) | 3.56 (1.19) | 3.58 (1.14) | 3.30 (1.26) |
| info about indep groups accessing linked info | 3.62 (1.07) | 3.50 (1.14) | 3.45 (1.16) | 3.48 (1.10) | 3.36 (1.19) |
| info about indep group reproductions | 4.02 (1.01) | 3.92 (1.04) | 3.75 (1.11) | 3.71 (1.03) | 3.70 (1.13) |
| info about indep robustness checks | 3.94 (1.04) | 3.80 (1.04) | 3.71 (1.10) | 3.68 (1.09) | 3.71 (1.12) |

**Figure 8.** Responses to preprint credibility questions by academic career stage. Mean and standard deviation of response to preprint credibility questions by career stage. Respondents who either skipped the question or listed a professional job title that could not be placed in the academic career ladder above were not included in this table. The response scale is 1—not at all important, 2—slightly important, 3—moderately important, 4—very important, 5—extremely important.

broader research population. The variation by career stage and discipline was rather weak, suggesting that the top-line conclusions of our findings may be generalizable. However, the significant underrepresentation of non-Western researchers is a significant limitation for making any inferences to those research communities.

Researchers were our main population of interest for this study, but they are not the only potential consumers of preprints; funders, journalists, policy makers, and the general public could all potentially benefit from preprints. For example, during the COVID-19 pandemic, major news sources used preprints regularly to keep the engaged public informed about the latest evidence. The Pew survey [22] described above indicates that there may be at least some consistency between what the public and researchers might value, and a recent preprint has called for more open science behaviours in COVID-19 preprints to increase rigor and the ability to assess rigor [39]. But more work is needed to understand if researchers and others generally favour similar or different cues on preprints.

We investigated researchers' self-reported opinions about the importance of various cues for assessing research credibility, but not their behaviour towards preprints. There is substantial evidence that self-reported intentions and attitudes are not always aligned with their behaviours [40,41]. Participants may have reported the beliefs about which cues should be important for assessing credibility, or what they think they would use when assessing credibility, but it is not clear whether these would be the most predictive of researchers' actual likelihood to read, believe, and cite papers. Therefore, it is important to directly measure the relative impact of these cues on behaviours. For example, would the presence of transparency and independent verification cues overwhelm the impact of author identity and usage cues on readers' likelihood to engage with a preprint? This goes far beyond what the present study can address. Nevertheless, even if the survey data represents only

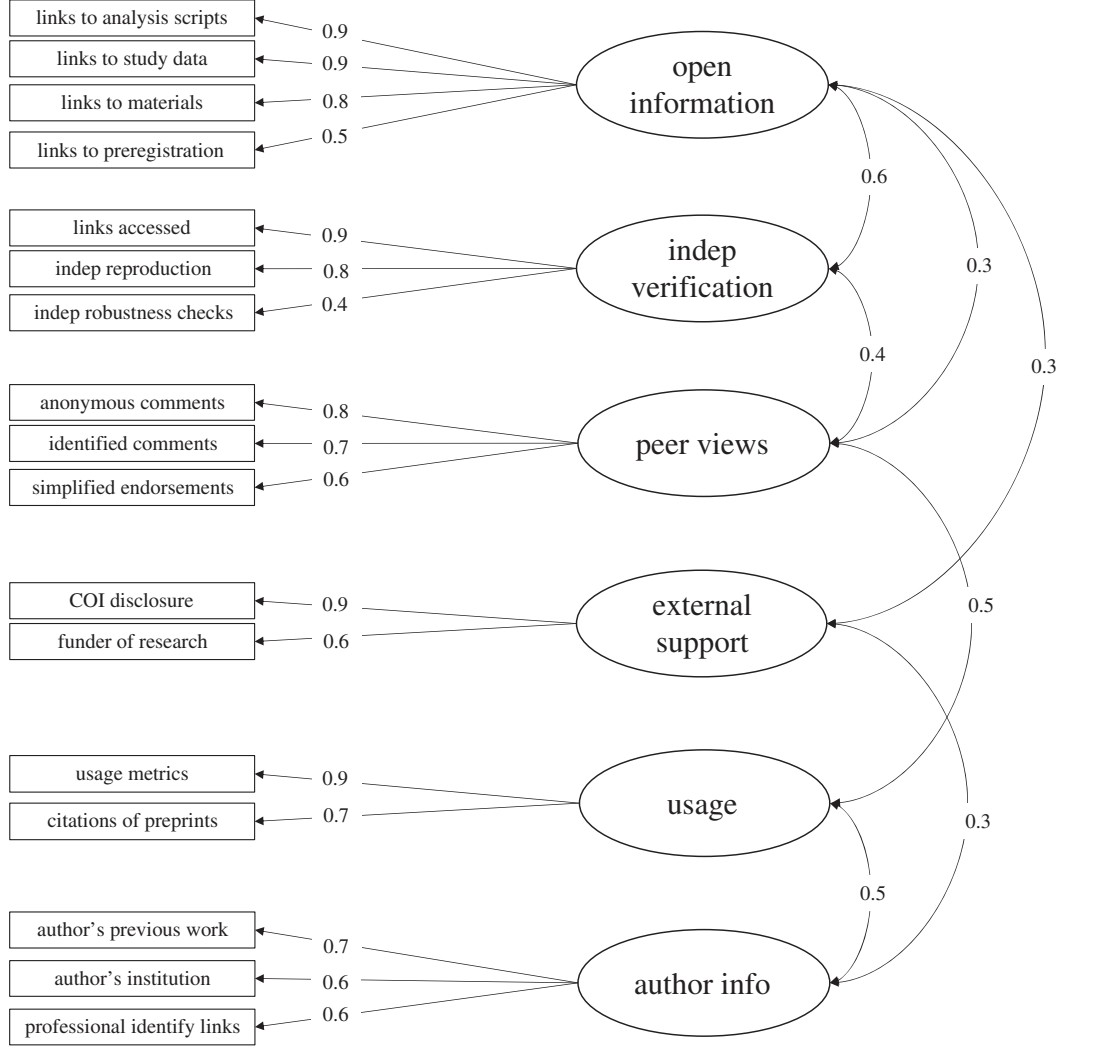

**Figure 9.** Factor structure of information items. The six-factor structure resulting from the EFA. Two items, 'authors general level of open scholarship' and 'preprint submitted to a journal', did not load onto any factor particularly strongly, and were not retained in further analyses.

researchers' ideals for credibility cues, this insight is important for informing the design and implementation of services to help researchers behave according to their ideals and to reduce the impact of unwanted biases.

## 3.2. Conclusion

We observed broad agreement among researchers from different disciplines and career stages that cues on preprints related to open content and independent verification of author claims would be very important for judging the credibility of preprints. It will be useful to replicate and extend our findings to other research communities and other potential cues for assessing research credibility. Open content and independent verification cues are rarely displayed by preprint services, so there is a big opportunity for services to add cues valued by researchers and potentially increase the credibility of preprints.

# 4. Methods

## 4.1. Materials

We drafted items starting with information that is commonly displayed on published work, initiatives related to best practices in research reporting (e.g. badges for open practices, Transparency and

| cues on preprint services | arXiv | SSRN | OSFpreprints | ChemRxiv | bioRxiv | preprints.org | PeerJ | NBER |
|---|---|---|---|---|---|---|---|---|
| author's previous work | no | no | no | no | no | no | no | no |
| author's institution | no | yes | no | partly[1] | yes | no | yes | no |
| professional identity links | no | no | no | partly[1] | yes | yes | no | no |
| COI disclosures | no | no | no | yes | no | no | yes | no |
| author's level of open scholarship | no | no | no | no | no | no | no | no |
| funders of research | no | no | no | no | no | no | yes | no |
| preprint submitted to a journal | no[2] | no[2] | no[2] | no[2] | no[2] | no[2] | no[2] | no |
| usage metrics | no | yes | yes | yes | no | yes | yes | no |
| citations of preprints | yes | yes | no | yes | no | no | yes | no |
| link to study data | no | n o | no[3] | no[3] | partly[4] | no | yes | no |
| link to study analysis scripts | no | no | no[3] | no[3] | partly[4] | no | no[3] | no |
| link to materials | no | no | no[3] | no[3] | no | no | no[3] | no |
| link to pre-reg | no | no | no[3] | no[3] | no | no | no[3] | no |
| independent groups accessing links | no | no | no | no | no | no | no | no |
| independent group reproductions | no | no | no | no | no | no | no | no |
| independent robustness checks | no | no | no | no | no | no | no | no |
| anonymous comments | no | no | partly[5] | no | partly[5] | partly[5] | no | no |
| identified comments | no | no | partly[5] | no | partly[5] | partly[5] | yes | no |
| simplified endorsements | no | no | no | no | no | no | no | no |

[1] service only shows this information for author who uploaded preprint

[2] service shows which pre prints have been accepted to journals, but not which submitted

[3] service has a general place to link/upload other files, but types of files are not clearly identified

[4] service has a specific location for data/code information, but does not differentiate between the two

[5] service require commenters to have a public username, but username doesn't have to be a real name

**Figure 10.** Information cues presented by preprint services. Coding by preprint service for whether they cue the information discussed in each preprint credibility question.

Openness Promotion [TOP] guidelines), and cues that had been identified during literature review as important for trust/credibility judgements. Then, we invited members of the OSF preprint services community, and other preprint stakeholders to revise the survey and assess clarity and pertinence for researchers from a broad variety of fields. Following this round of refinement, we pilot tested with researchers from a few disciplines to gather additional feedback on the clarity of the questions and the adequacy of the response options.

The final survey included questions in four categories: engagement information, importance of cues for credibility, credibility of service characteristics, and demographics (see https://osf.io/4qs68/ for the full version of the questionnaire [42]).

### 4.1.1. Engagement information

Four items asked participants about their familiarity with preprints, the extent to which they favour the use of preprints in their discipline, and how often (if ever) they viewed/downloaded preprints, and how often (if ever) they or one of their co-authors has posted a preprint. Because the word 'preprint' is not universal in all disciplines (e.g. 'working papers' are often posted before publication in economics)

and some disciplines use 'preprint' broadly to include both preprints and postprints, we defined 'preprint', 'postprint', and 'preprint service' for our participants.

### 4.1.2. Importance of cues for credibility

We asked participants to rate 19 different types of information for how important it would be to have them when assessing the credibility of a preprint. Some types of information (e.g. author[s] institutional information, links to available data) are presently listed on at least some preprint services (figure 10). However, other types of information (e.g. simplified endorsements of preprints by other researchers, information about whether independent groups could reproduce the findings in the preprint) were not displayed by any services at the time of the survey. The response scale was a 5-point Likert scale from 1 (*not at all important*) to 5 (*extremely important*).

### 4.1.3. Credibility of service characteristics

We asked participants the extent to which 18 characteristics would increase or decrease the credibility of the service as a whole. Some features related to specific functionality that could be provided by a preprint service (e.g. services screen for spam), while others related to behaviours users could engage in on the service (e.g. service allows the posting of new versions of preprints). The response scale was a 7-point Likert scale from −3 (*decrease a lot*) to 3 (*increase a lot)*. Analysis of this data can be found in the electronic supplementary materials.

### 4.1.4. Demographic information

We asked participants five questions to gather demographic information. These included the country they currently reside in, their job title/position, age, discipline, sub-discipline, and how they heard about the survey.

## 4.2. Procedure

We collected survey responses from late March 2019 to the end of June 2019. To investigate potential differences between discipline, career stage, and geographical location, we collected as diverse a sample as possible. We partnered with groups from different research communities, preprint services, publishers, scholarly societies, and individuals to diversify our outreach. Outreach included social media posts, emails to various listservs, emails to journals or platform lists, pop-up windows on journal pages, and some direct emails to departments.

Following consent, we defined 'preprints', 'preprint service', and 'postprint' for all respondents, and then asked them the preprint engagement questions. This was followed by the importance of cues for credibility section. During this section, we defined 'data', 'materials', 'pre-registration', 'replication', and 'reproduction' for respondents, as these terms are used differently in different disciplines and may have been unfamiliar to some respondents. They then responded to the service credibility questions, followed by the demographics. For discipline, respondents could self-select one of 29 disciplines that we knew to have preprint services or could select 'other'. They were then asked to enter their sub-discipline or discipline as free response text, and we used this information to code them into the three tiers of the bepress taxonomy of disciplines [43]. If respondents put a discipline that could not be cleanly coded into a single discipline at a given level of the bepress taxonomy, we treated the data as missing at that level.

Ethics. The protocol for the survey was approved by the IRB at the University of Virginia (Protocol Number: 2192). Informed consent was obtained from all participants in the first page of the survey.

Data accessibility. The data, materials, and code underlying this paper can be found on OSF: https://osf.io/6kz2j/.

Authors' contributions. C.K.S. collaborated on the design of the study, drafted the survey, programmed and administered the survey, carried out the statistical analyses and visualizations, and drafted the manuscript. T.M.E. and B.A.N. acquired funding, collaborated on the design of the study, provided critical feedback on the survey and provided critical feedback on the manuscript draft.

Competing interests. C.S., T.E. and B.N. are paid employees of the non-profit Center for Open Science that has a mission to increase openness, integrity and reproducibility of research including offering services supporting preprints.

Funding. This research was funded by a grant from the Alfred P. Sloan Foundation (Grant Number: G-2018-11108).

Acknowledgements. We had many individuals and groups assist us with survey drafting and/or recruitment. We would like to thank Oya Y. Rieger from Ithaka S+R (formerly from arXiv), Peter Binfield from PeerJ, Jessica Polka from

ASAPbio, Naomi Penfold from eLife (formerly from ASAPbio), Darla Henderson from ACS/ChemRxiv, Jon Tennant, Katherine Hoeberling from BITSS, Philip Cohen from SocArXiv, Wendy Hasenkamp from the Mind and Life Institute, Tom Narock from EarthArXiv, and Vicky Steeves from LISSA for helping to draft and refine survey items. We would also like to thank ASAPbio, PLOS, BITSS, FABSS, EACR, AMSPC, The Electrochemical Society, Lindau Alumni Network, F1000Research, Springer Nature, BMJ, SREE, Elsevier, Cambridge University Press, Wiley, eLife, iTHRIV, preprints.org, HRA, preLight, APA, APS, the Psychonomic Society, SocArXiv, PaleoRxiv, LISSA, EarthRxiv and Inarxiv for their help with survey recruitment.

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
