## [Reviewer comments · Royal Society Open Science]

Review History

RSOS-201520.R0 (Original submission)

Review form: Reviewer 1 (Todd Ross-Hellauer)

Is the manuscript scientifically sound in its present form?

Yes

Are the interpretations and conclusions justified by the results?

Yes

Is the language acceptable?

Yes

Do you have any ethical concerns with this paper?

No

Have you any concerns about statistical analyses in this paper?

No

Recommendation?

Accept as is

Comments to the Author(s)

This paper presents the results of a survey of 3,759 researchers across a wide range of disciplines to determine the importance of different cues for assessing the credibility of individual preprints and preprint services. The article is, in my opinion, ready to publish as is (save one typo I found - Line 244 - "whether they or a co-authors had ..." - co-author should be singular). The study is very well structured and clear, and although the sample can be accused of having a western-bias, the article does a very good job of describing the ways in which this might affect the ways in which the data can be interpreted.

Overall, the article is important in examining the ways in which signals of credibility of preprints are perceived by researchers, and so aids in the evidence-based implementation of Open Science. One potential point on which the authors might consider briefly expanding is at lines 432-435 where the authors point out that although cues for independent verification are considered highly relevant for credibility by researchers, this is seemingly very difficult to implement and requires more work. Here, the authors could expand briefly on which steps, in their view, might be required for funders and services to work together to build capabilities to gather and display this type of information - or what the basic building blocks of such could be. But this is just nit-picking and if the authors do not see the need for this, it is no reason to delay publication.

Review form: Reviewer 2

Is the manuscript scientifically sound in its present form?

Yes

Are the interpretations and conclusions justified by the results?

Yes

Is the language acceptable?

Yes

Do you have any ethical concerns with this paper?

No

Have you any concerns about statistical analyses in this paper?

No

Recommendation?

Accept with minor revision (please list in comments)

Comments to the Author(s)

This manuscript tackles the credibility of preprints and presents the data collected from a survey conducted in 2019. The manuscript is well written and very interesting. To the best of my knowledge it is the first survey of this kind and therefore the manuscript presents new findings. This is also a very timely topic given the surge of preprints during the COVID-19 pandemic [A,B].

I will mention first a couple of papers that I would argue the authors should mention in their manuscript and I will then continue with my comments on the manuscript in no particular order.

One manuscript that is very relevant to the work submitted by the authors is available with [A], mentioning that preprints were misused during the pandemic in particular by the media (this related to one of the points mentioned in the discussions by the authors also), and that preprints by their nature contained findings that should be taken with caution. I would tend to think that this could very introduce the very nice work presented by the authors in this manuscript.

Preprints are now screened more closely since the COVID-19 pandemic before being made available on the servers [B]

I found this statement particularly interesting and a useful take-away from the manuscript “In general, it appears that more ‘traditional’ metrics (e.g., author information, peer-review information) are negatively correlated with preprint views/downloads and favorability, indicating that these items were seen as more important by those lower on these scales, while the open science and independent- verification indicators tended to be rated as more important by those who tended to favor and use preprints. In general, the preprint submission variable did not correlate as strongly with the open science or independent verification indicators as either preprint viewing/downloading behavior or favorability towards preprints.” Although I suspected this already before reading the manuscript, having data to back up this “feeling” is better.

I am not familiar at all to exploratory factor analysis and cannot comment on it or the conclusion made from it. As such, I recommended that the paper be looked at by a statistician in case none of the other reviewers had enough expertise either.

“Even accepting that the peer review process is unreliable [9-12]” I would argue that here a quick sum up of why it’s unreliable and the references would be nice to have.

“Previous work found that displaying download count information about papers alongside abstracts altered download behaviors [25].” While most of the rest of the related work is presented with rather cautious words (using a lot of hedges such as “might” “could” etc...), this one is quite a strong statement especially if we consider that only one reference is given I would try to make it less assertive here, or perhaps justify why the authors do not use cautious language here by summarizing why the study they cite make the statement so strong. To be cautious, I would simply write, following what the authors have done so far “can alter download behaviour [25]”.

It is the first time I review for this journal so I am not sure whether or not linking all figures at the end of the submission is the de-facto standard or if it part of the guidelines, but having figures along the text that described them or discussed them would have made the reading of this submission much easier.

On that topic I also found a bit odd to have “Fig X” in the title of sub-sections.

Again, this might be a particularity of the format, but navigating subsections was a little bit difficult sometimes. Some figures were mentioned in bold after their subparts had been discussed (e.g., line 249 “Fig 4.A” and line 255 “Fig 4”).

There seems to be a typo line 370 “The 6-factor structure resulting from the EFA.” does not make sense to me.

This finding “...information,’ most do not differentiate between different types of information or cue particular types of content and few services enable preprint authors to provide links if relevant material...” is particularly interesting and calls for a change in the way preprint servers provide meta-data as early as possible. In the meantime, this findings might call for new “structured abstracts” that could contain reproducibility statements directly in the preprint.

“Researchers were our main population of interest for this study, but they are not the only 455 potential consumers of preprints; funders, journalists, policy makers, and the general public could all potentially benefit from preprints.” I would argue that this again calls for discussing what has recently been observed during COVID-19 with the potential misuse of preprints [A]. Could journalists better identify if a preprint is reliable with such additional information?

“There is substantial evidence that self-reported intentions and attitudes are not always aligned with their behaviors” was an important part of the discussions that I would have raised if the authors had not mentioned it and I am pleased to see that the authors identified this as a limitation of their study.

The acknowledgements are presented twice in the submission line 540 and line 30.

Of course I particularly appreciate that the authors linked to their materials for the study. I went through it to check whether some questions I had when reading the manuscript had their answers in these additional materials but could not always find an answer. Please note that these are just things I came to think about while reading the manuscript and not necessarily calling for a specific discussion or mention in the final version of the paper, although it might be interesting in some cases. For instance:

How do the authors define Conflicts of Interest (COIs) and do they think that respondents had different things in mind. The preprint I mentioned earlier [A] for instance analyzed conflicts of interests with the editorial board which are rarely mentioned. Do the authors think that any respondent could have had this in mind too?

Following up on this doesn't "Funder of the research" and declaration of COIs somehow overlap? Considering that both categories exhibit quite similar answers I do not think that this matters too much, but I would argue that often declaration of COIs basically comes to stating that the research was funded by a specific group. At least, these two categories are heavily related to me.

As it seems that most of my concerns can be addressed with a simple writing pass, I would therefore recommend that the manuscript be accepted with minor revisions.

I once again want to highlight the importance of the work presented here and I do hope that the authors can consider my few comments to strengthen the submission. I am looking forward to seeing this manuscript published.

REFS:

[A] <https://doi.org/10.1101/2020.08.13.249847>

[B] <https://www.nature.com/articles/d41586-020-01394-6>

Decision letter (RSOS-201520.R0)

Dear Dr Soderberg

On behalf of the Editors, we are pleased to inform you that your Manuscript RSOS-201520 "Credibility of preprints: An interdisciplinary survey of researchers" has been accepted for publication in Royal Society Open Science subject to minor revision in accordance with the referees' reports. Please find the referees' comments along with any feedback from the Editors below my signature.

Please submit your revised manuscript and required files (see below) no later than 7 days from today's (ie 24-Sep-2020) date. Note: the ScholarOne system will 'lock' if submission of the revision is attempted 7 or more days after the deadline. If you do not think you will be able to meet this deadline please contact the editorial office immediately.

Best regards,

on behalf of Professor Essi Viding (Subject Editor)
openscience@royalsociety.org

Associate Editor Comments to Author:

Thanks for this work, which two referees have commented on. The referees are broadly supportive of publication but have a number of recommendations that will help clarify or re-emphasise a number of points in your work. Please address these, and we'll look forward to receiving your revision in the near future.

Reviewer comments to Author:

Reviewer: 1
Comments to the Author(s)

This paper presents the results of a survey of 3,759 researchers across a wide range of disciplines to determine the importance of different cues for assessing the credibility of individual preprints and preprint services. The article is, in my opinion, ready to publish as is (save one typo I found - Line 244 - "whether they or a co-authors had ..." - co-author should be singular). The study is very well structured and clear, and although the sample can be accused of having a western-bias, the article does a very good job of describing the ways in which this might affect the ways in which the data can be interpreted.

Overall, the article is important in examining the ways in which signals of credibility of preprints are perceived by researchers, and so aids in the evidence-based implementation of Open Science. One potential point on which the authors might consider briefly expanding is at lines 432-435 where the authors point out that although cues for independent verification are considered highly relevant for credibility by researchers, this is seemingly very difficult to implement and requires more work. Here, the authors could expand briefly on which steps, in their view, might be required for funders and services to work together to build capabilities to gather and display this type of information - or what the basic building blocks of such could be. But this is just nit-picking and if the authors do not see the need for this, it is no reason to delay publication.

Reviewer: 2

Comments to the Author(s)

This manuscript tackles the credibility of preprints and presents the data collected from a survey conducted in 2019. The manuscript is well written and very interesting. To the best of my knowledge it is the first survey of this kind and therefore the manuscript presents new findings. This is also a very timely topic given the surge of preprints during the COVID-19 pandemic [A,B].

I will mention first a couple of papers that I would argue the authors should mention in their manuscript and I will then continue with my comments on the manuscript in no particular order.

One manuscript that is very relevant to the work submitted by the authors is available with [A], mentioning that preprints were misused during the pandemic in particular by the media (this related to one of the points mentioned in the discussions by the authors also), and that preprints by their nature contained findings that should be taken with caution. I would tend to think that this could very introduce the very nice work presented by the authors in this manuscript.

Preprints are now screened more closely since the COVID-19 pandemic before being made available on the servers [B]

I found this statement particularly interesting and a useful take-away from the manuscript "In general, it appears that more 'traditional' metrics (e.g., author information, peer-review information) are negatively correlated with preprint views/downloads and favorability, indicating that these items were seen as more important by those lower on these scales, while the open science and independent- verification indicators tended to be rated as more important by those who tended to favor and use preprints. In general, the preprint submission variable did not correlate as strongly with the open science or independent verification indicators as either preprint viewing/downloading behavior or favorability towards preprints." Although I suspected this already before reading the manuscript, having data to back up this "feeling" is better.

I am not familiar at all to exploratory factor analysis and cannot comment on it or the conclusion made from it. As such, I recommended that the paper be looked at by a statistician in case none of the other reviewers had enough expertise either.

"Even accepting that the peer review process is unreliable [9-12]" I would argue that here a quick sum up of why it's unreliable and the references would be nice to have.

"Previous work found that displaying download count information about papers alongside abstracts altered download behaviors [25]." While most of the rest of the related work is presented with rather cautious words (using a lot of hedges such as "might" "could" etc...), this one is quite a strong statement especially if we consider that only one reference is given I would try to make it less assertive here, or perhaps justify why the authors do not use cautious language here by summarizing why the study they cite make the statement so strong. To be cautious, I would simply write, following what the authors have done so far "can alter download behaviour [25]".

It is the first time I review for this journal so I am not sure whether or not linking all figures at the end of the submission is the de-facto standard or if it part of the guidelines, but having figures along the text that described them or discussed them would have made the reading of this submission much easier.

On that topic I also found a bit odd to have "Fig X" in the title of sub-sections.

Again, this might be a particularity of the format, but navigating subsections was a little bit difficult sometimes. Some figures were mentioned in bold after their subparts had been discussed (e.g., line 249 "Fig 4.A" and line 255 "Fig 4").

There seems to be a typo line 370 “The 6-factor structure resulting from the EFA.” does not make sense to me.

This finding “...information,’ most do not differentiate between different types of information or cue particular types of content and few services enable preprint authors to provide links if relevant material...” is particularly interesting and calls for a change in the way preprint servers provide meta-data as early as possible. In the meantime, this findings might call for new “structured abstracts” that could contain reproducibility statements directly in the preprint.

“Researchers were our main population of interest for this study, but they are not the only 455 potential consumers of preprints; funders, journalists, policy makers, and the general public could all potentially benefit from preprints.” I would argue that this again calls for discussing what has recently been observed during COVID-19 with the potential misuse of preprints [A]. Could journalists better identify if a preprint is reliable with such additional information?

“There is substantial evidence that self-reported intentions and attitudes are not always aligned with their behaviors” was an important part of the discussions that I would have raised if the authors had not mentioned it and I am pleased to see that the authors identified this as a limitation of their study.

The acknowledgements are presented twice in the submission line 540 and line 30.

Of course I particularly appreciate that the authors linked to their materials for the study. I went through it to check whether some questions i had when reading the manuscript had their answers in these additional materials but could not always find an answer. Please note that these are just things I came to think about while reading the manuscript and not necessarily calling for a specific discussion or mention in the final version of the paper, although it might be interesting in some cases. For instance:

How do the authors define Conflicts of Interest (COIs) and do they think that respondents had different things in mind. The preprint I mentioned earlier [A] for instance analyzed conflicts of interests with the editorial board which are rarely mentioned. Do the authors think that any respondent could have had this in mind too?

Following up on this doesn’t “Funder of the research” and declaration of COIs somehow overlap? Considering that both categories exhibit quite similar answers I do not think that this matters too much, but I would argue that often declaration of COIs basically comes to stating that the research was funded by a specific group. At least, these two categories are heavily related to me.

As it seems that most of my concerns can be addressed with a simple writing pass, I would therefore recommend that the manuscript be accepted with minor revisions.

I once again want to highlight the importance of the work presented here and I do hope that the authors can consider my few comments to strengthen the submission. I am looking forward to seeing this manuscript published.

REFS:

[A] <https://doi.org/10.1101/2020.08.13.249847>

[B] <https://www.nature.com/articles/d41586-020-01394-6>

===PREPARING YOUR MANUSCRIPT===

Your revised paper should include the changes requested by the referees and Editors of your manuscript. You should provide two versions of this manuscript and both versions must be provided in an editable format:one version identifying all the changes that have been made (for instance, in coloured highlight, in bold text, or tracked changes);a 'clean'

version of the new manuscript that incorporates the changes made, but does not highlight them. This version will be used for typesetting.

===PREPARING YOUR REVISION IN SCHOLARONE===

-- Ensure that your data access statement meets the requirements at <https://royalsociety.org/journals/authors/author-guidelines/#data>. You should ensure that you cite the dataset in your reference list. If you have deposited data etc in the Dryad repository, please only include the 'For publication' link at this stage. You should remove the 'For review' link.

Author's Response to Decision Letter for (RSOS-201520.R0)

See Appendix A.

Decision letter (RSOS-201520.R1)

Dear Dr Soderberg,

It is a pleasure to accept your manuscript entitled "Credibility of preprints: An interdisciplinary survey of researchers" in its current form for publication in Royal Society Open Science.

Due to rapid publication and an extremely tight schedule, if comments are not received, your paper may experience a delay in publication. Royal Society Open Science operates under a continuous publication model. Your article will be published straight into the next open issue and this will be the final version of the paper. As such, it can be cited immediately by other researchers. As the issue version of your paper will be the only version to be published I would

advise you to check your proofs thoroughly as changes cannot be made once the paper is published.

on behalf of Prof Essi Viding (Subject Editor)
openscience@royalsociety.org

Appendix A

We thank the reviewers and editor for helpful feedback on our manuscript. Below we have included a point by point response to each of the requests for edits made by the reviews.

Associate Editor Comments to Author:

Thanks for this work, which two referees have commented on. The referees are broadly supportive of publication but have a number of a recommendations that will help clarify or re-emphasise a number of points in your work. Please address these, and we'll look forward to receiving your revision in the near future.

Reviewer comments to Author:

Reviewer: 1 Comments to the Author(s)

This paper presents the results of a survey of 3,759 researchers across a wide range of disciplines to determine the importance of different cues for assessing the credibility of individual preprints and preprint services. The article is, in my opinion, ready to publish as is (save one typo I found - Line 244 - "whether they or a co-authors had ..." - co-author should be singular).

We have fixed this typo.

The study is very well structured and clear, and although the sample can be accused of having a western-bias, the article does a very good job of describing the ways in which this might affect the ways in which the data can be interpreted.

Overall, the article is important in examining the ways in which signals of credibility of preprints are perceived by researchers, and so aids in the evidence-based implementation of Open Science. One potential point on which the authors might consider briefly expanding is at lines 432-435 where the authors point out that although cues for independent verification are considered highly relevant for credibility by researchers, this is seemingly very difficult to implement and requires more work. Here, the authors could expand briefly on which steps, in their view, might be required for funders and services to work together to build capabilities to gather and display this type of information - or what the basic building blocks of such could be. But this is just nit-picking and if the authors do not see the need for this, it is no reason to delay publication.

We have added to this paragraph to highlight some of the specific blockers to implementing verification cues, and a few different ways funders and services could work to tackle these obstacles.

Reviewer: 2 Comments to the Author(s)

This manuscript tackles the credibility of preprints and presents the data collected from a survey conducted in 2019. The manuscript is well written and very interesting. To the best of my knowledge it is the first survey of this kind and therefore the manuscript presents new findings. This is also a very timely topic given the surge of preprints during the COVID-19 pandemic [A,B].

I will mention first a couple of papers that I would argue the authors should mention in their manuscript and I will then continue with my comments on the manuscript in no particular order.

One manuscript that is very relevant to the work submitted by the authors is available with [A], mentioning that preprints were misused during the pandemic in particular by the media (this related to one of the points mentioned in the discussions by the authors also), and that preprints by their nature contained findings that should be taken with caution. I would tend to think that this could very introduce the very nice work presented by the authors in this manuscript.

We have added a short mention of this citation to our manuscript, specifically where we discuss preprints by non-researchers (e.g. journalists).

Preprints are now screened more closely since the COVID-19 pandemic before being made available on the servers [B]

We have not added this citation to our manuscript. The additional screening process discussed in this article is mostly restricted to covid-preprints and only applies to some specific services. Because our manuscript is focused on preprints in general, rather than preprints on one particular research topic, we did not want to make it too covid-specific.

I found this statement particularly interesting and a useful take-away from the manuscript “In general, it appears that more ‘traditional’ metrics (e.g., author information, peer-review information) are negatively correlated with preprint views/downloads and favorability, indicating that these items were seen as more important by those lower on these scales, while the open science and independent- verification indicators tended to be rated as more important by those who tended to favor and use preprints. In general, the preprint submission variable did not correlate as strongly with the open science or independent verification indicators as either preprint viewing/downloading behavior or favorability towards preprints.” Although I suspected this already before reading the manuscript, having data to back up this “feeling” is better.

I am not familiar at all to exploratory factor analysis and cannot comment on it or the conclusion made from it. As such, I recommended that the paper be looked at by a statistician in case none of the other reviewers had enough expertise either.

“Even accepting that the peer review process is unreliable [9-12]” I would argue that here a quick sum up of why it’s unreliable and the references would be nice to have.

We have altered this section of the manuscript to give a short summary of the references that speak to the unreliability of the peer-review process.

“Previous work found that displaying download count information about papers alongside abstracts altered download behaviors [25].” While most of the rest of the related work is presented with rather cautious words (using a lot of hedges such as “might” “could” etc...), this one is quite a strong statement especially if we consider that only one reference is given I would try to make it less assertive here, or perhaps justify why the authors do not use cautious language here by

summarizing why the study they cite make the statement so strong. To be cautious, I would simply write, following what the authors have done so far “can alter download behaviour [25]”.

We have tweaked the wording of this section to soften this statement.

It is the first time I review for this journal so I am not sure whether or not linking all figures at the end of the submission is the de-facto standard or if it part of the guidelines, but having figures along the text that described them or discussed them would have made the reading of this submission much easier.

On that topic I also found a bit odd to have “Fig X” in the title of sub-sections.

Again, this might be a particularity of the format, but navigating subsections was a little bit difficult sometimes. Some figures were mentioned in bold after their subparts had been discussed (e.g., line 249 “Fig 4.A” and line 255 “Fig 4”).

In our revision, we are following the RSOS formatting guidelines.

There seems to be a typo line 370 “The 6-factor structure resulting from the EFA.” does not make sense to me.

This is note a typo, it is a description of what figure 10 shows.

This finding “...information,’ most do not differentiate between different types of information or cue particular types of content and few services enable preprint authors to provide links if relevant material...” is particularly interesting and calls for a change in the way preprint servers provide meta-data as early as possible. In the meantime, this findings might call for new “structured abstracts” that could contain reproducibility statements directly in the preprint.

“Researchers were our main population of interest for this study, but they are not the only 455 potential consumers of preprints; funders, journalists, policy makers, and the general public could all potentially benefit from preprints.” I would argue that this again calls for discussing what has recently been observed during COVID-19 with the potential misuse of preprints [A]. Could journalists better identify if a preprint is reliable with such additional information?

We have added to this section of the text to more specifically call out how journalists have used preprints during the pandemic (and citing article [A] mentioned by the reviewer), and how open science cues could be applicable to them as well.

“There is substantial evidence that self-reported intentions and attitudes are not always aligned with their behaviors“ was an important part of the discussions that I would have raised if the authors had not mentioned it and I am pleased to see that the authors identified this as a limitation of their study.

The acknowledgements are presented twice in the submission line 540 and line 30.

In our revision, we are following the RSOS formatting guidelines.

Of course I particularly appreciate that the authors linked to their materials for the study. I went through it to check whether some questions I had when reading the manuscript had their answers in these additional materials but could not always find an answer. Please note that these are just things I came to think about while reading the manuscript and not necessarily calling for a specific discussion or mention in the final version of the paper, although it might be interesting in some cases. For instance:

How do the authors define Conflicts of Interest (COIs) and do they think that respondents had different things in mind. The preprint I mentioned earlier [A] for instance analyzed conflicts of interests with the editorial board which are rarely mentioned. Do the authors think that any respondent could have had this in mind too?

Following up on this doesn't "Funder of the research" and declaration of COIs somehow overlap? Considering that both categories exhibit quite similar answers I do not think that this matters too much, but I would argue that often declaration of COIs basically comes to stating that the research was funded by a specific group. At least, these two categories are heavily related to me.

We did not define COIs for respondents, so it is difficult to know exactly what information respondents expect to be in a COI (e.g. whether they were thinking about editorial COIs or not). Behaviors and norms around COIs vary by field, and so potentially some respondents may have been thinking of this and others not. We decided not to include a further discussion of this in the revision, as it seemed like a minor point.

As it seems that most of my concerns can be addressed with a simple writing pass, I would therefore recommend that the manuscript be accepted with minor revisions.

I once again want to highlight the importance of the work presented here and I do hope that the authors can consider my few comments to strengthen the submission. I am looking forward to seeing this manuscript published.

REFS:

[A] <https://doi.org/10.1101/2020.08.13.249847>

[B] <https://www.nature.com/articles/d41586-020-01394-6>